# Genome-Wide Association Study of Potential Meat Quality Trait Loci in Ducks

**DOI:** 10.3390/genes13060986

**Published:** 2022-05-31

**Authors:** Qixin Guo, Lan Huang, Hao Bai, Zhixiu Wang, Yulin Bi, Guohong Chen, Yong Jiang, Guobin Chang

**Affiliations:** 1College of Animal Science, Technology of Yangzhou University, Yangzhou 225009, China; dx120190114@yzu.edu.cn (Q.G.); dx120200133@stu.yzu.edu.cn (L.H.); wangzx@yzu.edu.cn (Z.W.); ylbi@yzu.edu.cn (Y.B.); jiangyong@yzu.edu.cn (Y.J.); 2Joint International Research Laboratory of Agriculture and Agri-Product Safety, The Ministry of Education of China, Institutes of Agricultural Science and Technology Development, Yangzhou University, Yangzhou 225009, China; bhowen1027@yzu.edu.cn (H.B.); ghchen2019@yzu.edu.cn (G.C.)

**Keywords:** duck, meat quality, genome-wide associate study, single nucleotide polymorphism, copy number variants

## Abstract

With continuously increasing living standards and health requirements of consumers, meat quality is becoming an important consideration while buying meat products. To date, no genome-wide association study (GWAS) for copy number variants (CNVs) and single nucleotide polymorphisms (SNPs) has been conducted to reveal the genetic effects on meat quality in ducks. This study analyzed the phenotypic correlation and heritability of fat, water, collagen, and protein content of duck breast muscle. To identify the candidate variants for meat quality, we performed a GWAS using 273 ducks from an F2 population. The results of the SNP GWAS showed that the *BARHL2*, *COPS7B*, and *CCDC50* genes were associated with fat content; *BLM*, *WDR76*, and *EOMES* with water content; *CAMTA1*, *FGD5*, *GRM7*, and *RAPGEF5* with collagen production; and *RIMS2*, *HNRNPU*, and *SPTBN1* with protein content. Additionally, 3, 7, 1, and 3 CNVs were associated with fat, water, collagen, and protein content, respectively, in duck breast muscle. The genes identified in this study can serve as markers for meat quality. Furthermore, our findings may help devise effective breeding plans and selection strategies to improve meat quality.

## 1. Introduction

Over the last several decades, meat consumption has increased substantially worldwide. Customers’ expectations of meat safety and quality are increasing [1,2,3,4]. Meat quality is a complex trait that has several defining aspects, such as collagen, water, protein, and fat content, which play critical roles in consumer acceptance and product pricing [5,6,7]. Several studies on the meat quality of ducks, especially the fat, protein, collagen, and water content of breast muscle, have been conducted [8,9]. Numerous studies have shown that the fat content of meat is positively correlated with the tenderness of the meat, and increased fat content may reduce the physical strength of adipose tissue and connective tissue, resulting in improved meat sensory quality [10]. Fat is an important flavor precursor, and adipose tissue undergoes an oxidation reaction when heated to produce aromatic substances, which are the main source of duck meat flavor. The tenderizing ability of muscle depends on the interaction between muscle protein molecules. It is generally believed that muscles with high collagen content have high tenderness, and muscles with low collagen content have low tenderness. The main factor that affects muscle tenderness is thermally dissolved collagen. Meat quality, similar to other traits, is co-regulated by several factors related to environmental conditions and genetic background [11,12,13]. However, almost all meat quality traits have been shown to be potentially heritable [14,15]. The measurement of meat quality requires a lot of time and money, and so the improvement of breeding efficiency is lower. Therefore, using genomic variation (such as SNP and CNV, etc.) to predict the breeding value of meat quality is a promising strategy to improve meat quality. However, it is unclear which genome-wide SNPs and CNVs significantly affect duck meat quality.

The availability of genome-wide single nucleotide polymorphism (SNP) and copy number variant (CNV) panels has made genomic prediction possible for a broad range of livestock [16,17]. In the past several years, genome-wide association studies (GWAS) have identified an increasing number of potential markers associated with phenotypes of interest, including weight, plumage color [18,19,20,21], disease [22,23,24], and other economically important traits [25,26], thereby contributing to an extensive understanding of traits biology and providing a list of positional candidate genes for quantitative traits in poultry and livestock. Meanwhile, the GWAS has also revealed candidate sites and genes for meat quality, which is a critical economic trait for livestock and poultry species. In addition, functional analyses combined with GWASs have identified that a cluster of interacting genes and pathways are co-regulated with meat quality traits [27]. Previous studies have reported that ryanodine receptor 1 (*RYR1*), phosphorylase kinase catalytic subunit γ 1 (*PHKG1*), protein kinase AMP-activated non-catalytic subunit γ 3 (*PRKAG3*), melanocortin 4 receptor (*MC4R*), and insulin-like growth factor 2 (*IGF2*) are associated with meat quality in livestock [28,29,30,31]. Recently, epigenetics has also identified polymorphisms in seven epigenetic-related genes associated with meat quality in beef populations [32]. In addition, TYRO3 protein tyrosine kinase (*TYRO3*), microsomal Glutathione S-Transferase 1 (*MGST1*), kinesin family member 2A (*KIF2A*), and nucleoside-Triphosphatase cancer-related (*NTPCR*) were shown to be related to the intramuscular fat content in the breast muscle (IMF_Br_). Furthermore, the expression of the other genes examined (collagen type XII α 1 chain (*COL12A1*), vacuolar protein sorting 4 homolog B (*VPS4B*), BR serine/threonine kinase 2 (*BRSK2*), and forkhead box C1 (*FOXC1*)) was significantly lower in chickens with high abdominal fat weight (AbFW) and as a percentage of eviscerated weight (AbFP), but the magnitude of the difference was less [33].

In the present study, to identify the candidate loci affecting meat quality, based on fat, protein, collagen, and water content in breast muscle, 273 duck offspring were used for GWAS analysis. The results will allow farmers to gain insights into the underlying biological processes and identify potential genes and markers that might be utilized in breeding programs to increase the meat quality of ducks.

## 2. Materials and Methods

### 2.1. Ethics Statement

All blood samples were collected and meat quality traits were measured strictly in accordance with the guidelines proposed by the China Council on Animal Care and Ministry of Agriculture of the People’s Republic of China. The study was approved by the Institutional Animal Care and Use Committee and the School of Animal Experiments Ethics Committee (license number: SYXK (Su) IACUC 2012-0029), Yangzhou University.

### 2.2. Breeding Experiments or Breeding for Sample Size

The experimental population consisted of 273 F_2_ segregating ducks. To construct the F_2_ segregating population, the F_1_ generation was produced from orthogonal crosses between the Chinese Crested (CC) ducks and Cherry Valley (CV) ducks. In the orthogonal experiment, 86 CC and 13 CV ducks were randomly selected and divided into 7 families. The number of offspring in the F_1_ generation exceeded 500 individuals. Moreover, there were no ducks with crest traits in the F_1_ generation. The ratio of male to female ducks was consistent. The F_2_ generation was produced from the natural mating of F_1_ hybrids, and mating was internally limited to orthogonal experiments. In building the families, we considered and complied with the following principles: (1) the male-to-female ratio was 1:3, (2) males and females in the same family were not from the same nest, and (3) female ducks within a family were not half-siblings. The F_2_ generation was composed of almost 2000 ducks that displayed segregation of various genetic characteristics, including meat quality-related traits. After the ducklings hatched, they were weighed on a weekly basis. Three weeks after hatching, all members of the F_2_ generation were moved from the duckling house to a designed individual shed and raised to the age of six weeks. We performed a slaughter experiment with more than 800 ducks and measured a series of traits, including meat quality. In all families, we found that the color trait followed the recessive inheritance of Mendel’s law of segregation. To identify candidate loci for meat quality-related phenotypes, we randomly selected 21 ducks per lineage (273 in total) from 800 individuals that had been measured for a series of traits for genome resequencing.

### 2.3. Meat Quality Trait Detection and Correlation Analysis

All meat quality measurements were taken on the left side of the carcasses. To increase the accuracy of meat quality detection results, we first removed the fascia from the breast meat and cut it into small pieces. Thereafter, we put the small piece of meat into a high-speed universal crusher to make a puree. Finally, FoodScan™ (FOSS, Hillerød, Denmark) was used to measure the crude protein content, crude fat content, water content, and collagen content [34]. To determine the correlation of all meat qualities, the *R*/corrplot package was used for the phenotypic correlation visualization [35].

### 2.4. Variant Calling and Genotyping

For GWAS analysis, 273 samples were aligned to the genome of mallard (assembly number: GCA_008746955.1) using BWA (settings: mem-t 4-k 32-M-R) [36]. The sample alignment rates were 96–98%. The average coverage depth for the reference genome (excluding the N region) was between 9.34–15.74X, and the 4X base coverage (≥4) was greater than 82.64%. Variant calling was performed for all samples using the Genome Analysis Toolkit (GATK) v 3.7, with the UnifiedGenotyper method [37]. The SNPs were filtered using the Perl script. After filtering, the GWAS sample retained 12.6 Mb of SNPs (filter conditions: only two alleles; single-sample quality = 5; single-sample depth = 5–75; total-sample quality = 20; total-sample depth = 273–1,000,000; maximum missing rate of individuals and site = 0.1; and a minor allele frequency (MAF) = 0.05).

### 2.5. CNV Calling

CNVnator merged the BAM files with a bin size of 200 bp [38]. After CNV calling, quality control was performed on the raw CNV data for each duck. The parameters of filtering were a *p*-value < 0.01 (pval1 calculated using *t*-test statistics), size > 1 kb, and q0 < 0.5. A *p*-value <0.01 indicated that the region between two calls was not the same CNV, and q0 was the fraction of mapped reads with zero quality. In addition, CNVs that overlapped with gaps or unplaced chromosomes (chrUn in the crested duck genome) were removed.

### 2.6. Population Structure and SNP and CNV Distribution

PLINK was used to identify the underlying population structure and principal component analysis (PCA) was performed within and across all 273 samples. *R*/ggplot2 was used for PCA visualization. The *R*/CMplot package was used to visualize SNP and CNV distributions on chromosomes [39].

### 2.7. Genome-Wide Association Analysis

Association analyses were conducted using a linear mixed model (LMM) to correct for the population structure and kinship matrix using EMMAX [40]. Briefly, the general mixed model used in this approach can be specified as: y=Xβ+Zμ+e, where y represents an n × 1 vector of phenotypes, X is an n × q matrix of fixed effects, β is a q × 1 vector representing the coefficients of fixed effects, and Z is an n × t matrix relating the random effect to the phenotypes of interest.

Manhattan plots illustrating the GWAS results were produced using the *R*/qqman package in *R* [41]. The significance threshold (α) of the association of SNP and CNV markers with different traits was calculated using the Bonferroni correction.

## 3. Results

### 3.1. Meat Quality Trait Correlation Analysis

The descriptive statistics for all meat quality traits in the F_2_ population of ducks used in the present study are shown in Table 1. Based on descriptive statistics analysis, the maximum and minimum values of fat content were 26.42% and 20.52%, respectively. The coefficient of variation (CV) of protein (21%) was higher than that of collagen (16%). To identify candidate variants for all meat quality traits, we first performed a normal test on all meat quality traits. The results show that all meat quality traits are normally distributed (Figure 1a). Moreover, through the normal distribution test of the meat quality traits of male ducks and female ducks, it was found that they all followed a normal distribution (Figure 1b). Through the correlation between the traits, it was found that there was a very significant negative correlation between fat and protein and water, respectively (Figure 1c). In addition, the phenotypic heritability (*h*^2^) of water, fat, protein, and collagen content was 0.2454, 0.5310, 0.401, and 0.42, respectively.

### 3.2. SNP Disequilibrium and Population Structure

After SNP quality control, 12,661,915 SNPs were distributed in across 39 chromosomes. Chromosome 1 showed the highest number of SNPs, whereas chromosome 25 contained the fewest SNPs. The MAF for all SNPs was re-calculated after quality control; only a MAF of >5% was retained. The PCA of the population was used to investigate the genetic structure and relationship. The PCA results showed that the first three PCs explained 11.99%, 10.24%, and 6.11% of the total genetic variation, respectively (Figure 2a). The population was divided into six separate candidate clusters, demonstrating potential stratification in the reference population. The distribution of the SNP information within 1 Mb windows on different chromosomes is shown in Figure 2b.

### 3.3. GWAS for Four Meat Quality Traits Based on SNPs

The efficient mixed-model association eXpedited (EMMAX) was used to identify candidate SNPs of four meat quality traits: fat, collagen, protein, and water content in the breast muscle of ducks. The results of GWAS showed that 41 SNPs located on chromosomes 8 and 10 were associated with breast muscle fat content (Figure 3a, Appendix A). Nearest to the candidate SNPs for fat content, the following genes are located: barh like homeobox 2 (*BARHL2*, APL8), COP9 signalosome subunit 7b (*COPS7B*) and Coiled-Coil domain containing 50 (*CCDC50*, APL10). The three SNPs nearest to the BLM RecQ Like Helicase (*BLM,* APL2), WD repeat domain 76 (*WDR76,* APL8), and eomesodermin (*EOMES*) genes were associated with water content in breast muscle (Figure 3b, Appendix A). Three SNPs located on chromosome 22, one SNP located on chromosome 11, and one SNP located on chromosome 9 were candidates associated with collagen content and were located nearest to the genes calmodulin binding transcription activator 1 (*CAMTA1,* APL22), fyve, rhogef, and ph domain containing 5 (*FGD5,* APL11), glutamate metabotropic receptor 7 (*GRM7*), and rap guanine nucleotide exchange factor 5 (*RAPGEF5,* APL22), respectively (Figure 3c, Appendix A). In addition, six candidate SNPs located on chromosomes 2, 3 and 34 were associated with the protein content of breast muscle. The protein content-associated SNPs were nearest to the genes of regulating synaptic membrane exocytosis 2 (*RIMS2,* APL2), heterogeneous nuclear ribonucleoprotein u (*HNRNPU,* APL3), and spectrin β, non-erythrocytic 1 (*SPTBN1,* APL34) (Figure 3d, Appendix A).

### 3.4. GWAS for Four Meat Quality Traits Based on CNVs

In addition to SNPs, CNVs are another important type of genetic variation. A total of 4163 CNVs were extracted using CNVnator (Figure 4). In the present study, all CNVs and four meat quality traits were used for GWAS. Three CNVs nearest to the genes, *WASH* complex subunit 7 *(WASH7)*, GRB2 associated binding protein 1 *(GAB1)*, Transporter 1, ATP binding cassette subfamily b member (*TAP1*), and major histocompatibility complex, class II, dm β (*DMB*) were associated with fat content in the breast muscle (Figure 5a, Appendix A). Seven CNVs associated with water content were neatest to the following genes: collagen type XI α 1 chain (*COL11A1*), transporter 1, ATP binding cassette subfamily b member (*TAP1*), bromodomain containing 2 (*BRD2*), major histocompatibility complex, class II, dr α (*DRA*), complement component 4 (*C4*), lysine methyltransferase 2e (*KMT2E*), MLX Interacting Protein (*MLXIP*), and rb transcriptional corepressor 1 (*RB1*) (Figure 5b, Appendix A). A CNV nearest to coiled-coil serine rich protein 1 (*CCSER1*) was associated with collagen content (Figure 5c, Appendix A). Furthermore, three CNVs closest to the genes dedicator of cytokinesis 2 (*DOCK2*), inhibitory synaptic factor family member 2b (*INSYN2B*), PTTG1 Regulator of Sister Chromatid Separation, Securin (*PTTG1*), and Acyl-CoA Binding Domain Containing 6 (*ACBD6*) were associated with protein content (Figure 5d, Appendix A).

## 4. Discussion

In recent years, the demand for poultry has continuously increased. In addition, with changes in consumer demand, poultry and livestock farmers want to obtain high-quality meat individuals to adapt to changes in the consumer market. Meat quality is an important quality trait of livestock and poultry meat products that directly influences the production performance and economic situation of farms [42,43]. Thus, exploring and finding candidate association variants and genes for meat quality may improve breeding efficiency and the consumer’s desire to buy meat [12]. As suggested by several studies, the accuracy of GWAS and the heritability (*h*^2^) of traits have a close relationship. In this study, the *h*^2^ of fat, water, collagen, and protein content in duck breast muscle was found to be 0.531, 0.2454, 0.42, and 0.401, respectively, which was calculated based on the phenotype. Moreover, there was a significant negative correlation between fat, moisture, and protein content. A previous study of Australian lamb found that the *h*^2^ of intramuscular fat (IMF) content, pH_24_, and myoglobin content (Myo) were 0.44, 0.32, and 0.25, respectively [14]. Previous studies reported or indicated meat quality has important effects on the oxidative stability, tenderness, and juiciness of livestock and poultry meat. Some previous studies based on Illumina 60K also identified 2 SNPs in Beijing-You chicken, which were located nearest the Cholecystokinin (*CCK*) and toll interacting protein (*TOLLIP*) genes on chromosomes 2 and 5, respectively [44]. At the same time, in the GWAS analysis of Jinghai-Yellow chicken, it was found that cerebellin 2 precursor (*CBLN2*), *LOC101747478*, hematopoietic prostaglandin d synthase (*HPGDS*), SET Domain Containing 2, Histone Lysine Methyltransferase (*SETD2*), ankyrin repeat domain 46 (*ankrd46*), zinc finger protein, fog family member 2 (*ZFPM2*) and glutamate metabotropic receptor 4 (*GRM4*) genes were significantly correlated with IMF [45]. Furthermore, the ECM receptor interaction pathway was a significantly enriched IMF-related pathway, which was evidenced by the compared and analyzed transcriptome of Zhuanghe Dagu chicken and Arbor Acres chickens [46]. Meanwhile, some previous reports considered that IMF differs from other fats in three ways: metabolic activities, adipocyte size, and developmental timing. Moreover, non-muscular adipocytes are larger than intramuscular adipocytes in cattle and pigs [47,48]. In the present study, we detected 41 SNPs located on chromosomes 8 and 10 neatest to the *BARHL2*, *COPS7B*, and *CCDC50* genes. Meanwhile, three CNVs that were nearest to the genes *WASHC4, GAB1* and *TAP1*, were related to fat content. Of these genes, the expression pattern in humans demonstrates that *CCDC50* was shown to be associated with body fat in humans and mice [49].

For collagen content in muscle, collagen is an important component of meat quality. Collagen is an abundant connective tissue protein and is a factor in changes in meat tenderness and texture [50,51,52]. However, the disclosure of candidate genes for collagen is conducive to the selection and breeding of collagen for ducks and provides consumers with higher-quality meat products. In the present study, three SNPs nearest to *CAMTA1*, *FGD5*, *GRM7*, and *RAPGEF5* were identified as the candidate for collagen content in duck breast muscle. Moreover, a CNV nearest to *CCSER1* was associated with collagen content. The research in humans and mouse reported that FGD5 regulated the collagen IV content [53].

To date, relatively few GWAS have been performed on water and protein content in meat. In general, meat composition consists of approximately 75% of water, 19% of protein, 2.5% of fat, 1.2% of carbohydrates and 1.65% of nitrogen compounds [54]. Therefore, water and protein are important components of meat products. At the same time, the content of water and protein is directly related to the tenderness and water retention of meat. Therefore, the selection of water and protein content in meat can greatly improve meat quality. In the present study, three and six candidate SNPs were associated with water and collagen content, respectively. In addition, seven and three CNVs were associated with water and protein content in duck breast muscle. The identification of associated SNPs represents a key step forward in dissecting the genetic basis of meat quality-related traits in ducks and is also helpful for further demonstrating molecular mechanisms of related traits and designing better selection methods for these traits.

## 5. Conclusions

The results of the current study show that 41 SNPs and 3 CNVs potentially with fat content, 3 SNPs and 7 CNVs are candidates associated with water content, 3 SNPs and 1 CNV are candidates associated with collagen content, and 6 SNPs and 3 CNVs are candidates associated with protein content. These findings improve our understanding of poultry genetics and provide a genetic basis for breeding programs aimed at maximizing the economic potential of duck breeding and farming.

## Figures and Tables

**Figure 1 genes-13-00986-f001:**
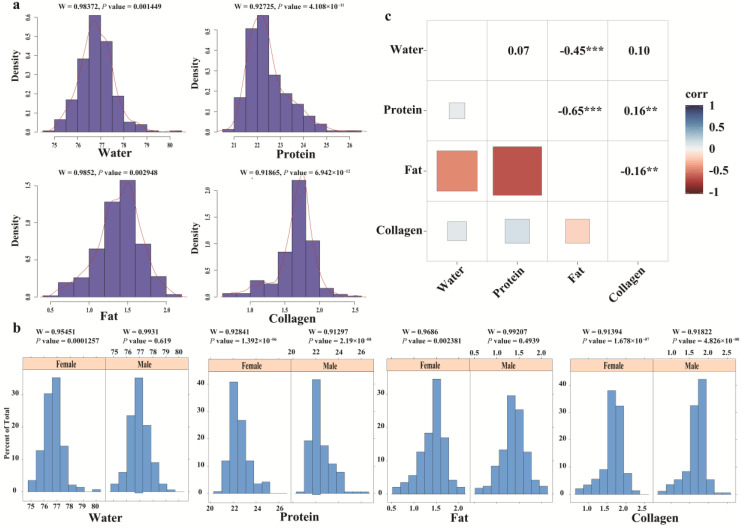
The phenotype of meat quality statistics analysis. (**a**) Frequency distribution of the adjusted phenotypes of meat quality; (**b**) frequency distribution of the adjusted phenotypes of male and female meat quality; (**c**) pairwise Pearson correlation coefficients for the four meat quality traits; ‘**’ represents the *p* value less than 0.01 (*p* value ≤ 0.01); ‘***’ represents the *p* value less than 0.001 (*p* value ≤ 0.001).

**Figure 2 genes-13-00986-f002:**
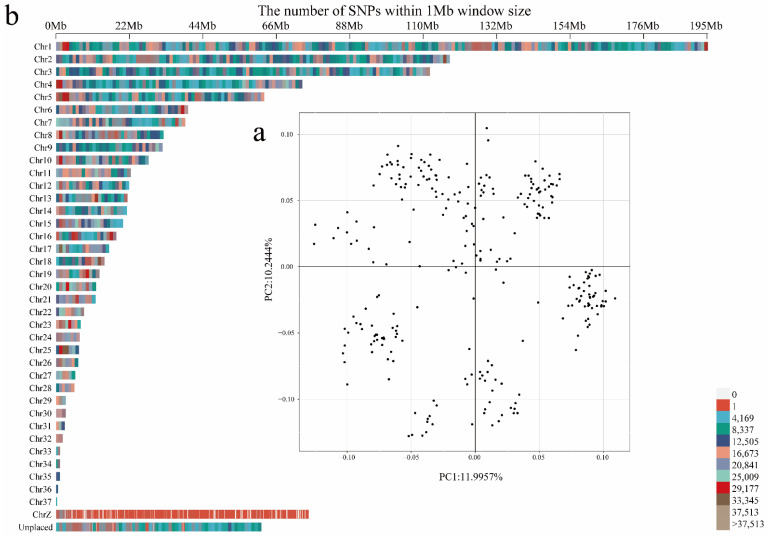
SNP distribution and PCA analysis of all samples of the present study. (**a**) PCA; (**b**) SNP distribution.

**Figure 3 genes-13-00986-f003:**
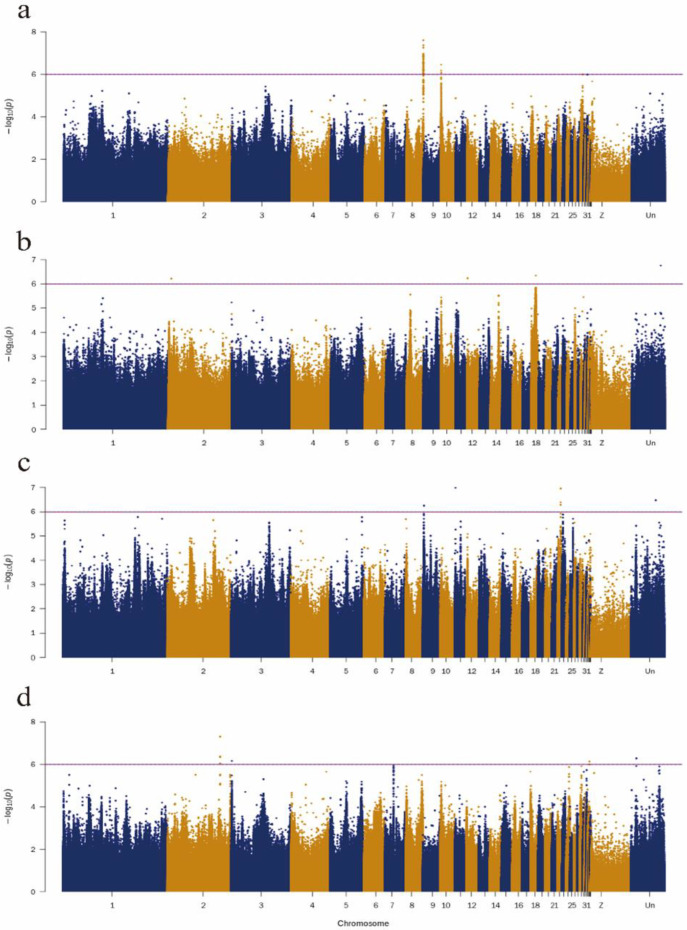
The Manhattan plots of meat quality traits. GWAS analysis for fat (**a**), water (**b**), collagen (**c**), and protein (**d**) content in breast muscle. The x-axis represents the chromosomes, and the y-axis represents the −log10 (*p*-value).

**Figure 4 genes-13-00986-f004:**
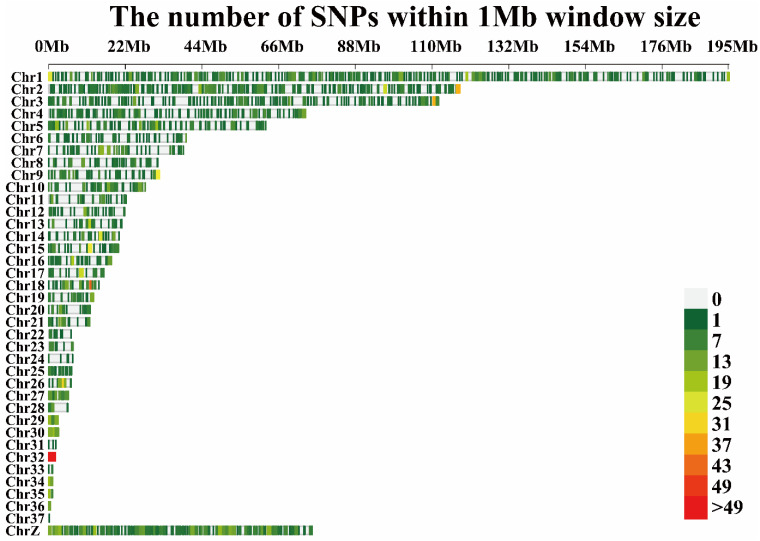
Distribution of CNVs on chromosomes.

**Figure 5 genes-13-00986-f005:**
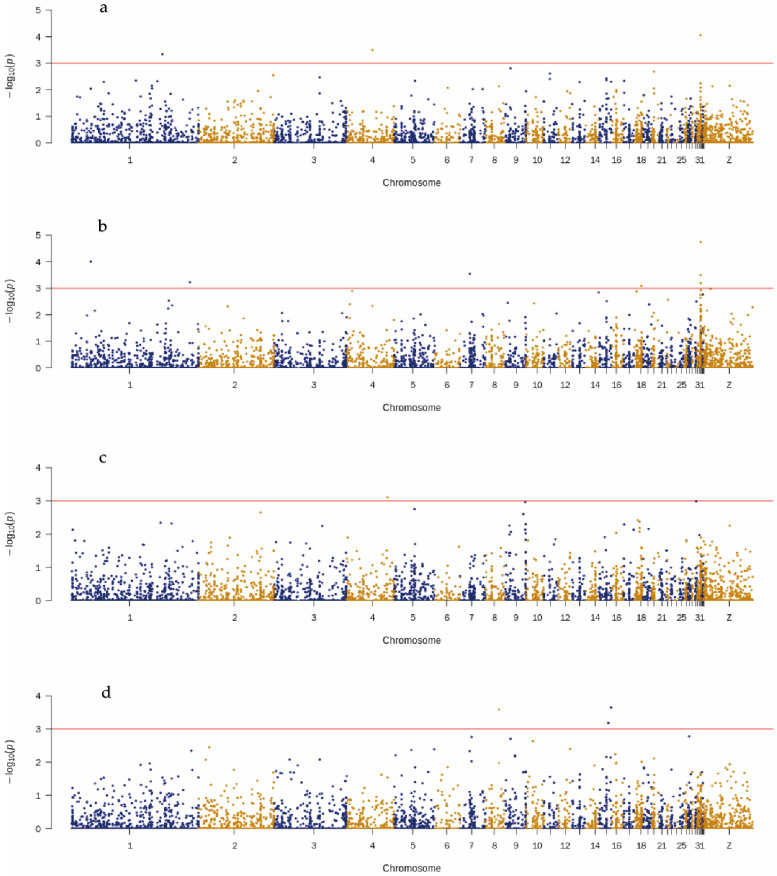
The Manhattan plots of meat quality traits. CNV-based GWAS analysis for fat (**a**), water (**b**), collagen (**c**), and protein (**d**) content in breast muscle. The x-axis represents the chromosomes, and the y-axis represents the −log10 (*p*-value).

**Table 1 genes-13-00986-t001:** Descriptive statistics of meat quality trait ^a^.

Trait	Mean (g)	SD (g)	CV	Min (g)	Max (g)
Water	76.83	0.76	0.01	74.98	80.12
Fat	22.47	0.9	0.04	20.52	26.42
Protein	1.39	0.29	0.21	0.58	2.07
Collagen	1.65	0.27	0.16	0.73	2.52

^a^ n = 273.

## Data Availability

The genome assembly and all of the re-sequencing data used in this research are deposited in the Genome Sequence Archive (GSA) at National Genomics Data Center (http://bigd.big.ac.cn/, 16 March 2022) Beijing Institute of Genomics, Chinese Academy of Sciences (GSA: CRA005019).

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
