# Peer review of "Genome-Wide Association Study of Potential Meat Quality Trait Loci in Ducks"

_genes, 2022, doi:10.3390/genes13060986_

Round 1

Reviewer 1 Report

The manuscript entitled " Genome-Wide Association Study for Screening and Identification of Potential Meat Quality Trait Loci in Ducks" is very interesting and could be applied in practice. Methods which were applied and statistical analysis are in area of current investigations.

The text, however must be improved before publish.

There are 3 main inaccuracies to change/explanation:

Material and methods

It is not clear how experimental animals were chosen; F1 was composed from 500 ducks, F2 - 2000. From these 2000 ducks 800 were slaughtered for traits analysis and than 273 were genotyped.

What guided the authors in reducing the number of genotyped animals from 2000 to 273? Please explain it!

Results

Figures should be described in text chronologically e.g. line 174 - Fig. 2b, line 178 - Fig. 1a.

Moreover, there are no links between text and all figures (only 1a, 2b, 3a, 3b, 3c, 3d, 4, 5a, 5b, 5c, 5d are mentioned in the text).

Discussion

I think it is weakest point of this paper and should be improved. Authors discussed their results only with one reference [11]. Almost half of the text is repeat of earlier text from Results section.

IT CAN NOT BE PUBLISHED IN THIS FORM!

Please find other papers about duck GWASs or other species GWASs (meat quality). Check what is the role of indicated candidate genes and compare them to other results (maybe someone found that these genes are associated with meat quality or other traits in different species?)...

There are also some smaller changes that should be applied:

line 14 - of on meat

line 16 - of in duck

line 17 - showed

lines 18-20 - "are associated" can be omitted

line 21 - respectively,

line 22- of for meat

I propose following keywords with order: duck, meat quality, GWAS, SNP, CNV (or full names)

line 30 - phenotype trait

line 33 - animal

lines 41-42 - economically important traits

line 39-43 - in which species?

line 42 - traits

line 45 - for in

lines 47, 48, 51, 53 - give full name of genes and add information about analyzed species

line 49 - on epigenetics

lines 53-55 - rewrite this sentence because is not clear

lines 55-57 - this sentence looks like unfinished, please mention in which context these traits were investigated

line 88 - separation segregation

line 134 - F2 F2

line 170 - "in 39 chromosomes" or "across 39 chromosomes"

line 172 - fewest lowest

Section 3.3

 "The genes nearest to the candidate SNPs of fat content were located on BARHL2 on chromosome 8 and COPS7B and CCDC50 on chromosome 10."

I think that "Nearest to the candidate SNPs for fat content following genes are located: BARHL2 (APL8), COPS7B and CCDC50 (APL10)." sounds better.

Please apply it for all similar sentences in this section.

Lines 232, 234 - delete "breast muscle"

Sections 3.3 and 3.4 - please give full names of genes

Lines 279-280 - give species name for this study

Line 283 - of for

GOOD LUCK!!!

Author Response

The manuscript entitled " Genome-Wide Association Study for Screening and Identification of Potential Meat Quality Trait Loci in Ducks" is very interesting and could be applied in practice. Methods which were applied and statistical analysis are in area of current investigations.

The text, however must be improved before publish.

There are 3 main inaccuracies to change/explanation:

Material and methods

It is not clear how experimental animals were chosen; F1 was composed from 500 ducks, F2 - 2000. From these 2000 ducks 800 were slaughtered for traits analysis and then 273 were genotyped.

What guided the authors in reducing the number of genotyped animals from 2000 to 273? Please explain it!

Response: Thank you for drawing our attention to this point. We apologize for the unclear or erroneous description. In the present study, the 13 families were established using the F1 hybrids. The F2 generation was composed of almost 2000 ducks. And then, we random screened more than 63 ducks from each family for phenotypic testing. Next, we random selected 21 ducks from each family for whole-genome resequencing.    

Results

Figures should be described in text chronologically e.g. line 174 - Fig. 2b, line 178 - Fig. 1a.

Response: Thank you for pointing this out. We have change and reorder it.  

Moreover, there are no links between text and all figures (only 1a, 2b, 3a, 3b, 3c, 3d, 4, 5a, 5b, 5c, 5d are mentioned in the text).

Response: Thank you for pointing this out. We have added and recheck the figure links text of all manuscript.   

Discussion

I think it is weakest point of this paper and should be improved. Authors discussed their results only with one reference [11]. Almost half of the text is repeat of earlier text from Results section.

IT CAN NOT BE PUBLISHED IN THIS FORM!

Please find other papers about duck GWASs or other species GWASs (meat quality). Check what is the role of indicated candidate genes and compare them to other results (maybe someone found that these genes are associated with meat quality or other traits in different species?)...

Response: Thank you for reviewing our manuscript and providing suggestions to improve it. We have re-write the part of discussion.

There are also some smaller changes that should be applied:

line 14 - of on meat

line 16 - of in duck

line 17 - showed

lines 18-20 - "are associated" can be omitted

line 21 - respectively,

line 22- of for meat

I propose following keywords with order: duck, meat quality, GWAS, SNP, CNV (or full names)

line 30 - phenotype trait

line 33 - animal

lines 41-42 - economically important traits

line 39-43 - in which species?

line 42 - traits

line 45 - for in

lines 47, 48, 51, 53 - give full name of genes and add information about analyzed species

line 49 - on epigenetics

lines 53-55 - rewrite this sentence because is not clear

lines 55-57 - this sentence looks like unfinished, please mention in which context these traits were investigated

line 88 - separation segregation

line 134 - F2 F2

line 170 - "in 39 chromosomes" or "across 39 chromosomes"

line 172 - fewest lowest

Section 3.3

 "The genes nearest to the candidate SNPs of fat content were located on BARHL2 on chromosome 8 and COPS7B and CCDC50 on chromosome 10."

I think that "Nearest to the candidate SNPs for fat content following genes are located: BARHL2 (APL8), COPS7B and CCDC50 (APL10)." sounds better.

Please apply it for all similar sentences in this section.

Lines 232, 234 - delete "breast muscle"

Sections 3.3 and 3.4 - please give full names of genes

Lines 279-280 - give species name for this study

Line 283 - of for

GOOD LUCK!!!

Response: Thank you for the comments. We have revised the manuscript accordingly and have to change the mistake base on your comments point by point.

Reviewer 2 Report

The manuscript initiates an interesting area of poultry (ducks) research. However, the research design and writing style are heavily flawed and as such doesn’t stand as a quality publication. Some critical comments are;

There is NO information about genomic data in the Abstract. It is not evident throughout the manuscript that the genomic data is actually based on sequencing.

Lines 30-31: May delete repeated use of ‘content’.

Lines 35-36: Need attention.

Lines 43-45: These are repeating the same information as in previous sentence.

Introduction does not provide any context on Duck meat production, and why it is important to study these traits in Ducks.

Lines 87-88: Some details of prevalent colors are required.

Line 101: “273 samples” of what? SNPchip based genotypes or whole genome sequencies, whichever was relevant should provide details.

Line 107: “12.6 Mb of SNPs” ?

Lines 107-113: Better put this information in a table.

Line 127:  change “mixed linear model”, to “linear mixed model” and don’t need to abbreviate if it never being used in the manuscript. Moreover, the model used should be expressed appropriately.

Line 128: provide full form of EMMAX here rather than later at line 190.

Line 130: what is α= ?

Line 136: “maximum and minimum values”, should be given as “minimum and maximum”. By the way, why Fat content are only described in text out of all 4 traits?

Line 137: What is “FI”?

Line 143: Where is Figure S1?

Line 178: Figure 1a?

It is really hard to understand all Figures given the quality. Also, it is important to keep the order of all traits same in all Tables and Figures.

Lines 270-299 (Discussion): As such it repeats everything from Introduction and Results and nothing is discussed to provide any insights with any reference to previous knowledge.

Author Response

The manuscript initiates an interesting area of poultry (ducks) research. However, the research design and writing style are heavily flawed and as such doesn’t stand as a quality publication. Some critical comments are;

There is NO information about genomic data in the Abstract. It is not evident throughout the manuscript that the genomic data is actually based on sequencing.

Response: Thanks for your comments, and we have added the information about genomic data in the Abstract.  

Lines 30-31: May delete repeated use of ‘content’.

Lines 35-36: Need attention.

Lines 43-45: These are repeating the same information as in previous sentence.

Introduction does not provide any context on Duck meat production, and why it is important to study these traits in Ducks.

Response: Thanks for your comments, and we have added the information about why the meat quality was important to study these traits in Ducks.

Lines 87-88: Some details of prevalent colors are required.

Line 101: “273 samples” of what? SNPchip based genotypes or whole genome sequencies, whichever was relevant should provide details.

Response: Thanks for your comments. The 273 samples were based on the whole-genomic sequence. And we have added the information on methods.

Line 107: “12.6 Mb of SNPs” ?

Lines 107-113: Better put this information in a table.

Line 127:  change “mixed linear model”, to “linear mixed model” and don’t need to abbreviate if it never being used in the manuscript. Moreover, the model used should be expressed appropriately.

Line 128: provide full form of EMMAX here rather than later at line 190.

Line 130: what is α= ?

Line 136: “maximum and minimum values”, should be given as “minimum and maximum”. By the way, why Fat content are only described in text out of all 4 traits?

Line 137: What is “FI”?

Line 143: Where is Figure S1?

Line 178: Figure 1a?

It is really hard to understand all Figures given the quality. Also, it is important to keep the order of all traits same in all Tables and Figures.

Lines 270-299 (Discussion): As such it repeats everything from Introduction and Results and nothing is discussed to provide any insights with any reference to previous knowledge.

Response: Thank you for the comments. We have revised the manuscript accordingly and have to change the mistake base on your comments point by point.

Round 2

Reviewer 1 Report

The text is improved and can be published in present form, however, there are some points that need correction:

  • Replace sentence:

"To identified the candidate variants of meat quality, we performed
a GWAS using the 273 ducks from F2 population genomic resequencing data."

by

"To identify the candidate variants for meat quality, we performed
a GWAS using the 273 ducks from F2 population." in Abstract

  • Line 30 - delete "phenotype"
  • Line 57 - delete "in"
  • Line 104 - delete "separation"
  • Line 217 - delete "chromosomes 8"
  • Give full name of genes in Discussion

Author Response

The text is correctly improved and can be published in present form, however, there are some points that need correction:

  • Replace sentence:

"To identified the candidate variants of meat quality, we performed
a GWAS using the 273 ducks from F2 population genomic resequencing data."

by

"To identify the candidate variants for meat quality, we performed
a GWAS using the 273 ducks from F2 population." in Abstract

  • Line 30 - delete "phenotype"
  • Line 57 - delete "in"
  • Line 104 - delete "separation"
  • Line 217 - delete "chromosomes 8"
  • Give full name of genes in Discussion

Response: Thank you for reviewing our manuscript and providing suggestions to improve it. We have revised the manuscript accordingly and have responded to your comments point by point. 

Reviewer 2 Report

In addition to the previously provided comments, it is strongly suggested that the manuscript should be professionally edited. There are so many contradictory statements and misquotations of results in discussion. A formal response to each comment and track changes of all edits are also required to make the final decision.

Author Response

In addition to the previously provided comments, it is strongly suggested that the manuscript should be professionally edited. There are so many contradictory statements and misquotations of results in discussion. A formal response to each comment and track changes of all edits are also required to make the final decision.

Response: Thanks for your important comments. According to your comment, we have revised the manuscript and change the mistake point by point. And thank you for this suggestion and we have carefully revised the manuscript to improve the language. Further, a professional English editing service was used to proofread the manuscript and make editorial improvements.
